# A New Bioassay Platform Design for the Discovery of Small Molecules with Anticancer Immunotherapeutic Activity

**DOI:** 10.3390/md18120604

**Published:** 2020-11-29

**Authors:** Carmela Gallo, Giusi Barra, Marisa Saponaro, Emiliano Manzo, Laura Fioretto, Marcello Ziaco, Genoveffa Nuzzo, Giuliana d’Ippolito, Raffaele De Palma, Angelo Fontana

**Affiliations:** 1Bio-Organic Chemistry Unit, CNR-Institute of Biomolecular Chemistry, Via Campi Flegrei 34, 80078 Naples, Italy; giusi_barra@hotmail.it (G.B.); m.saponaro@icb.cnr.it (M.S.); emanzo@icb.cnr.it (E.M.); nuzzo.genoveffa@icb.cnr.it (G.N.); gdippolito@icb.cnr.it (G.d.); raffaele.depalma@unige.it (R.D.P.); 2Consorzio Italbiotec, Via Fantoli, 16/15, 20138 Milan, Italy; l.fioretto@icb.cnr.it; 3BioSearch Srl., Villa Comunale c/o Stazione Zoologica “A.Dohrn”, 80121 Naples, Italy; m.ziaco@icb.cnr.it; 4Internal Medicine, Clinical Immunology and Translational Medicine, University of Genova and IRCCS-Hospital S. Martino, 16132 Genova, Italy; 5Department of Biology, University of Naples Federico II, Via Vicinale Cupa Cintia 21, 80126 Naples, Italy

**Keywords:** chemical immunology, immunotherapy, dendritic cell, screening guidelines, drug discovery, anticancer, high throughput, bioassay platform

## Abstract

Immunotherapy takes advantage of the immune system to prevent, control, and eliminate neoplastic cells. The research in the field has already led to major breakthroughs to treat cancer. In this work, we describe a platform that integrates in vitro bioassays to test the immune response and direct antitumor effects for the preclinical discovery of anticancer candidates. The platform relies on the use of dendritic cells that are professional antigen-presenting cells (APC) able to activate T cells and trigger a primary adaptive immune response. The experimental procedure is based on two phenotypic assays for the selection of chemical leads by both a panel of nine tumor cell lines and growth factor-dependent immature mouse dendritic cells (D1). The positive hits are then validated by a secondary test on human monocyte-derived dendritic cells (MoDCs). The aim of this approach is the selection of potential immunotherapeutic small molecules from natural extracts or chemical libraries.

## 1. Introduction

The immune system has long been known to patrol cancer surveillance but only in the last decade the role of the immune system has been strongly reconsidered to fight cancer [1,2]. Immunotherapeutic approaches aiming to enforce immune response against neoplastic cells and include adoptive cell therapies, anticancer vaccines, engineered T cells with chimeric antigen receptors (CAR-T), monoclonal antibodies, and immunomodulators have been more and more exploited. Recently, a major breakthrough in immunotherapy has been the introduction of checkpoint inhibitors to boost T cell response against cancer cells [3,4].

Molecules able of activating presentation by innate immune cells are considered a suitable tools to boost the physiological immune response against cancer [5]. Dendritic Cells (DCs) represent the most efficient professional antigen-presenting cells (APC) and, in a tumoral contest, they recognize, process, and present tumor-associated antigens (TAA) complexed to major histocompatibility complex (MHC) molecules. After antigen uptake, DCs undergo maturation, releasing cytokines and providing priming for naïve T cells, allowing their polarization and activation [6,7]. In cancer, this machinery is often hampered, due to the tumor microenvironment (TME).

The link between inflammation and tumor progression is already well known and identified as a critical hallmark of cancer. In the inflammatory TME, besides the activation of processes associated with cancer cell survival and proliferation, the ability of cancer cells to release cytokines and activate molecular pathways, as that of NF-kB or STAT-3, allows the subversion of the adaptive immune response, suppressing an effective antitumor immunity and inhibiting DC maturation [8]. Furthermore, cancer cells induce immune downregulation promoting polarization and recruitment of regulatory T cells (Treg), tumor-associated macrophages (TAMs), and myeloid-derived suppressor cells (MDSCs) [9]. Tumor cells can downregulate MHC and costimulatory molecules, thus evading immune recognition [10]. Furthermore, the neoplastic proliferation leads to a decrease in oxygen and pH, which alter the metabolism of normal immune cells [11,12,13]. DCs able to infiltrate tumor are so often directed towards a tolerogenic phenotype that can enhance tumor progression by favoring Treg activation and the anti-inflammatory microenvironment [14,15].

Drugs such as oxaliplatin or anthracyclins [16] can activate the immune response through the release of damage-associated molecular patterns (DAMPs) by cancer cells as, for instance, apoptotic bodies that interact with pattern recognition receptors (PRRs) expressed on DCs. This mechanism leads to an indirect anticancer activity through a process of regulated cell death known as immunogenic cell death (ICD) [17,18]. An emerging group of drugs, reported as promising therapy, is represented by small molecule-based immunomodulators; they are directed against TAMs and MDSCs decreasing their infiltration in the tumor microenvironment [19]. Recently, we have reported the immunomodulatory activity of a novel class of sulfoglycolipids collectively named Sulfavants [20]. These compounds trigger unconventional DC maturation and in vivo antigen-specific immunization [21]. The initiation of a systemic immune response by the stimulation of innate immune cells correlates to adjuvanticity and PRR-mediated signaling. Sulfavants are under preclinical trials as vaccine adjuvants, and their efficacy has been already proven in a murine model of vaccine against melanoma [21]. Interestingly, the product is not cytotoxic but treated mice do not show progress of the tumor for more than 10 days after subcutaneous injection of B16F10 melanoma cells. Preliminary evidences support that Sulfavants can induce DC activation to overcome immune tolerance (manuscript in preparation).

This result has prompted us to characterize other small molecules that are able to activate DCs and, potentially, enhance immune response against neoplastic cells. The aim of the present work is to describe how we can implement a screening platform to select natural products with potential application in cancer immunotherapy. In consideration of the role of the DC-T cell axis in the development of anti-cancer therapies, we propose an innovative approach based on the use of a growth factor-dependent immature mouse dendritic cell (D1) and a panel of tumor cell lines for the selection of primary fractions, followed by the validation on human monocyte-derived dendritic cells (MoDCs). 

## 2. Results 

### 2.1. Immunophenotypic Assay on Dendritic Cells

In order to develop a reproducible and sensitive in vitro assay, we used a cell line resembling DCs for the identification of new immunomodulatory compounds. We selected a well characterized growth factor–dependent immature DC line derived from mouse spleen (D1) [22,23,24,25] that can be maintained in laboratory for over one year. To test the transition from an “immature” to a “mature” status [7,26], cells were monitored for changes in the expression of MHC class II (MHC-II) and the costimulatory molecules CD80 and CD40 [27]. These factors are functional for an efficient activation of DC- T “immunological synapse”. We also evaluated cell viability by propidium iodide (PI) in order to verify that compounds are non-toxic. In this experimental contest, target compounds are defined as non-toxic when the viability is at least 80%. Only these samples were selected for more in-depth biochemical analysis and optimization.

D1 cells can be maintained in laboratory, but their growth is strictly dependent on the presence in the culture medium of GM-CSF and fibroblast supernatant [22]. These cells can be maintained in culture for a limited time after thawing, thus we tested the optimum passage number (P) to perform the screening assay. Along 12 days, cells were split five times (P5), and cells at P3 and P4 from six different batches displayed the lowest and most constant basal levels of MHC-II, CD80, and CD40 (Figure 1a). D1 at P3 and P4 were then stimulated with LPS at the conditions previously reported [24] (10 µg/mL; *n* = 5) (Figure 1b) and a significative and reproducible upregulation of all surface markers were observed. By performing treatments at serial dilutions of these cells with LPS at 24 h, the optimal cell number for the assay resulted to be 1.5 × 10^5^ cell/0.2 mL of medium in a 96-well plate. Analogously, the treatment of P3 and P4 D1 cells with Sulfavant A [21] in the range from 1 to 60 µg/mL gave a dose-response maturation curve with the major effect on markers at 30 µg/mL in seven replicates (*n* = 7) (Figure 1c). Tumor necrosis factor-α (TNF-α) as an established inflammatory product following D1 cell activation was also measured as an additional condition (Appendix A). 

For the assays, the serial dilution of chemical fractions required the selection of solvents that do not affect response and viability of D1 cells. DMSO is commonly used to solubilize chemical products in cell tests, but preliminary results pointed out that this solvent was toxic to D1 at a concentration of 0.5% (1 µL in 0.2 mL assay volume) and interfered in cell maturation at lower concentrations. On the other hand, MeOH resulted in being toxic when added to the solution, but it did not affect cell vitality if we performed plate coating. Thus, mixtures of natural products and pure molecules were diluted in MeOH at the maximum concentration of 0.3 mg/mL and 0.05 mL of this solution was added to each test well. Plates were then left 3 h at room temperature to dry out and then used for the assay.

### 2.2. Cytotoxicity Assay on Tumor Cell Lines

After the design of the D1 assay, our work focused on assembling a panel of cancer cells for the cytotoxic tests. As proof of concept, we selected nine different cell lines (Table 1) between chronic forms of lung carcinoma (LC), melanoma (Mel), and multiple myeloma (MM) [28]. These cells have alterations in genes considered as “hot spots” in cancer for their aggressivity and frequency in the population. The lung cancer line HCC827 has a typical EGFR tyrosine kinase domain deletion (E746-A750) that is associated with a reduced survival, frequent lymph node metastasis, and poor chemosensitivity [29,30]. In addition, a recent study related this particular deletion to the repression of antitumor immunity mediated by DCs [31]. CALU-1 cells are mutated for Kras and p53, concurrent gene mutations linked to a major incidence of distant metastasis [32], while CALU-3 cells are only p53 mutated. CALU-1 Kras mutation (Gly12Cys), that is found in 11%–16% of lung adenocarcinomas (45–50% of mutant KRAS is G12C), appears of particular interest [33]. For Mel models, we used BRAF cell lines with Val600Glu mutation that is found in approximately 40%–60% of resistant melanoma in the Caucasian population [34] and is the target of the main chemotherapeutic treatments [35,36]. Among the three Mel cells, amelanocytic (A375 and A2058) and melanocytic (MALME 3M) lines were selected because of the different mechanisms of drug resistance [37,38]. For MM, the three cell lines are mutated in TP53 gene [39] and are representative of the three main different forms of this pathology, namely multiple myeloma (KMS-12), plasmacytoma (RPMI8226), and plasma cell leukemia (JJN-3). 

The nine cell lines showed different media and culture conditions. For this reason, the preliminary screening was performed using more than one protocol, paying attention to achieve easy, fast, and reproducible responses (data not shown). After an accurate titration of the cell number and the comparison of results obtained from several replicates, SRB assay was chosen for cells growing in adherence ((HCC827, CALU-1, CALU-3, A375, A2058), while MTS assays were chosen for cells in suspension (MALME-3M, KMS-12, RPMI8226, JJN-3). Cells were plated at a density of 1 × 10^4^ in 0.1 mL of medium in 96-wells, and DMSO was used as the solvent to dissolve chemical products at a maximum concentration of 1% (1 µL in 0.1 mL volume). 

Doxorubicin [40,41], MEK inhibitor [42,43], and cisplatin [44], which have a wide spectrum of cytotoxicity, were chosen as the positive controls.

### 2.3. Validation of the Hits Selected in D1 and Cytotoxic Assay

The ability to trigger an innate immune response by products that induce D1 maturation and do not show nonspecific cytotoxicity against the tumor cell lines is finally validated on a model of human primary cells that takes into account the physiological variability of donors (Figure 2). For the purpose of the screening platform, we selected human DCs. These cells are rare, accounting for approximately 0.1–1% of total blood mononuclear cells (PBMCs) and are difficult to maintain in culture. In alternatives to isolation methods, human DCs can be generated in vitro from peripheral blood CD14^+^ monocytes after magnetic activated cell sorting (MACS) and differentiation with GM-CSF and IL-4. The monocyte-derived DC (MoDCs) are well characterized and have long been used as a model to understand the biology of cross-presentation. Fully differentiated MoDCs efficiently present soluble and cell-associated antigens to T cells and have a stimulating capacity comparable to tissue-derived DCs [45]. Furthermore, the transition from immature to mature cells is easily assessed by measuring the expression of MHC-II (HLA-DR) and the co-stimulatory molecules CD83 and CD86 by flow cytometry.

The setting for the MoDCs assay was the same as D1. Cells were plated in 96-well plates at a density of 1.5 × 10^5^ cell/0.2 mL of medium and incubated for 24 h with stimuli after coating in MeOH. As the control of DC activation, the Toll-like receptor 2 (TLR-2) ligand Pam2CSK4 at a concentration of 1 µg/mL was used together with Sulfavant A in the range of 0.1 to 60 µg/mL [20]. The maturation of cells was assessed at 24 h by the measurement of human DC activation markers (Appendix A). In consideration of the biological variation of primary cells, the analysis was carried out on cells obtained by three different donors.

Positive “candidates” at this step were then monitored for diverse endpoints of MoDCs maturation, such as cytokine and chemokine analysis production via mRNA and protein quantification. Cytokines produced during DC activation influence the immune responses, generating subtypes of CD4^+^ T cells such as T helper 1 (Th1) cells, Th2 cells, regulatory T cells, or tumor-reactive CD8^+^ cytotoxic T cells [46].

### 2.4. Application of the Screening Platform to A Marine Natural Extract

To assess the selectivity of the screening methodology described above, we used a marine diatom, *Thalassiosira weissflogii* (CCMP-1336), as a case study. Molecules isolated from different marine diatoms exhibit antitumor and immunomodulatory functions [47,48]. It has been already reported that *T. weisfloggii* extract contains α-sulfoquinovosyl diacylglycerols (α-SQDGs) [21] and atypical phosphoglycolipids (PGDGs) [49] that induce the maturation of DCs in the micromolar range.

Methanol extracts (Ext) were obtained from the diatom wet pellet and fractionated by the Solid Phase Extraction (SPE) method that we have recently described [50]. Accordingly, organic extracts were suspended in water and loaded on a poly(styrene-divynylbenzene)-based support (HR-X). The major components of the mixture were then selectively eluted in order to give five fractions (A–E). An aliquot of 0.15 mg of the raw extract and enriched fractions B–E were tested at the concentrations of 5 and 30 µg/mL on both D1 cells and the panel of tumor cell lines in agreement with the procedures described above. Fraction C significantly upregulated the expression of CD80 and MHC-II on D1 cells in a dose-dependent manner (Figure 3a) with no relevant cytotoxicity on the tumor cell lines (Figure 3b). Except for the extract that showed a mild activity on CD80 expression, the other fractions showed no effects on both cell systems.

The fraction C was then tested on MoDCs at the concentration range between 0.01 and 30 µg/mL. As shown in Figure 4a, the expression of HLA-DR, CD86 and, at a lesser extent, CD83, were upregulated after treatment. According to a previous report on similar products [20], the expression of surface markers increased in a dose-dependent manner up to 10 µg/mL and slightly decreased at higher concentrations. NMR analysis confirmed the enrichment of α-SQDGs in fraction C (Appendix A), further stressing the strong predictive value of the proposed screening protocol for the selection of immunomodulatory small molecules. Fraction C was further purified by radial silica chromatography in eight fractions. Each fraction was tested again on MoDCs, and the activity matched the presence of α-SQDGs that were identified by NMR and by LC-MS/MS analysis, according to our recent report [51]. Figure 4b shows the increase of HLA-DR^+^ DCs along the fractionation steps. This trend is similar with the other maturation markers (not shown) and fits well with the purity degree of SGDGs in the active fraction.

## 3. Discussion

The last decade has brought significant successes to the understanding how the human immune system can be manipulated to fight cancer. In this context, the importance of an innate immune system has been highlighted. Cells of the innate immune system, like natural killer cells (NK), dendritic cells (DCs), macrophages, and mast cells have been recognized as active players in immunotherapy. Besides the classical ability to detect, target, and kill cancer cells, innate immunity has a fundamental role in shaping the cytokine and chemokine milieu of the TME, thus affecting activation and differentiation of effector T cells. DCs constitute a bridge between the innate and adaptive immune system that in TME can switch to a tolerogenic/immunosuppressive behavior and account for T cell priming. A deeper understanding of the mechanisms and factors controlling innate immune activation within TME should provide new therapeutic targets. Small molecules can either reduce immune suppression in the tumor milieu or enhance the activation of cytotoxic lymphocyte responses to the cancer [52,53]. The rationale for the pursuit of small molecule- based immune therapies is the broad range of cellular processes that can be targeted.

In this field, the potential anticancer and immunological activity of natural products represent an excellent and unexplored source for the development of new drugs. As far as we know, a defined procedure for the in vitro selection of small molecules for innate system activation combined with cytotoxic assay on tumor cell lines has not been reported yet.

The entire platform (Figure 5) relies on a two-step process based on DCs. In particular, the preliminary screening involves the use of a well-characterized model of mouse DCs (D1) that can be maintained in laboratory for long time [25], whereas the final validation is achieved by primary DCs derived from human monocytes (MoDCs) that are obtained from the discarded residues after the processing of donated blood. The combination of these two cellular systems has been designed to solve the main issues that have limited so far routinely use of DCs: the difficulty in purifying or maintaining in vitro functionality of these cells. In fact, DCs represent a specific tool to test the mechanisms of innate immune activation and, together with other cellular systems [54,55,56], have been already proposed for the discovery of immunomodulatory products [57,58]. Nowadays functional DC-like cells are obtained by treating murine or human myeloid monocytes with GM-CSF alone or in combination with growth factors [59,60,61]. Unfortunately, these cell lines can be propagated for no longer than three months, which limits their use in a screening process. Other options of DC substitutes for in vitro tests are genetically modified DC lines [62,63,64,65] or a few myeloid immortalized cells (Raw264.7 and J774, THP-1, HL-60 and MUTZ-3) [66,67]. These cell lineages can be maintained for long time and display a dendritic phenotype, but their functional and transcriptional profiles are only partially similar to those of human DCs [24].

The aim of the selection is to find small molecules like Sulfavant-A that can boost the innate immune response without being cytotoxic to cancer cells. In this approach, the nonspecific stimulation of the innate immune system by D1 assay is inversely correlated to a specific cytotoxicity on a panel of tumor cells. Like with Sulfavant, the selection of chemical candidates is carried out on the basis of the ratio between D1 activity and cytotoxicity, with the higher index being the better response.

The use of a wide panel of cancer cell lines and adequate immune cells models allow the experimental reproducibility that is a key factor to ensure a high throughput screening (HTS) and the selection of active compounds. As proof of concept, we showed that the designed screening platform achieved the identification of DC activators in a natural extract after fractionation by chromatographic means. This strategy has the advantages of removing interfering components, thus revealing the presence of minor active compounds and avoiding false positives. Results can be further investigated by in vitro MoDCs-T cells co-culture experiments to verify the outcome of the DC-T cell interaction, generating subsets of CD4^+^ T cells such as T helper 1 (Th1) cells, Th2 cells, regulatory T cells, or tumor-reactive CD8^+^ cytotoxic T cells.

In conclusion, this study provides a method for the identification of new small molecules that can target DC within the TME and potentially stimulate the immune response against cancer. Selected compounds are in fact able to induce functional DC maturation and could be exploited in different strategies for anticancer therapy such as inoculation of DC activators within the tumor, delivery of adjuvant/antigen carriers by nanoparticles directly targeting DCs, or administration of ex vivo generated, loaded-, or unloaded-DCs. The study also shows the opportunity of using chemical immunology approaches to explore small molecules for modulating immune response.

## 4. Materials and Methods

### 4.1. General

All spectrophotometric measures were performed at 4300 chromate plate reader (Awareness Technology). All FACS analysis were recorded at ACCURI C6 flow cytometer (Becton Dickinson). Mek inhibitor was from Bayer, cisplatin was from Accord Healthcare and doxorubicin was from Abcam; Lipopolysaccharides from E. coli (serotype 055:B5) was from Sigma-Aldrich. DMSO and MeOH used for dissolving the compounds were of HPLC grade (Sigma-Aldrich, Milan, Italy). All culture media and supplements were from Gibco (Thermo Fisher Scientific).

### 4.2. Cell Lines

Among the tumor cell lines: CALU-1, CALU-3, HCC827, MALME-3M, A375, A2058 were purchased from American Type Culture Collection (ATCC); the 3 lines of multiple myeloma, KMS-12, RPMI 8226, JJN-3 were purchased form the German Collection of Microorganisms and Cell Cultures (DSMZ).

D1 cell line was a gift of Professor Francesca Granucci from the University of Milano-Bicocca, Department of Biotechnology and Bioscience, Milan, Italy.

### 4.3. Cell Culture Conditions

CALU-1 were cultured in McCoy media supplemented with 10% FBS; CALU-3 in EMEM media completed with 10% FBS; HCC827 and RPMI8226 in RPMI with 10% FBS; MALME-3M in IMDM with 20% FBS; A375 and A2058 in DMEM 10% FBS; KMS-12 in RPMI 20% FBS; JJN3 in 40% IMDM 40% DMEM, 20% FBS.

D1 cells were maintained in IMDM supplemented with 30% and then in 15% R1-conditioned medium as described [22]. In previous reports, D1 cultures were supplemented with 30% fibroblast medium (NIH/3T3 SN) containing 10–20 ng/mL mouse rGM-CSF (R1 medium). We performed culture tests with a scalar dose of this medium from 1 to 20 ng/mL of mouse rGM-CSF and 5–10 ng/mL was selected as the best dose for the maintaining of D1 in culture without affecting the doubling rate (data not shown). These cells need to be maintained in non-treated culture dishes to avoid cell activation.

All culture media were supplemented with 1% of penicillin/streptomycin solution.

### 4.4. Cell Lines Titration and Treatments

Each tumor cell line was serially diluted (from 0.5 × 10^4^ to 2 × 10^4^) in 0.1 mL of medium and seeded in a 96-well plate. After 24 h cytotoxicity assays were performed. The best cell number for the assay for all cell lines resulted to be 1 × 10^4^ cells/ well, as this value is included in the linear range of the curve obtained plotting control absorbance versus cell number.

Cell lines were cultured at the concentration of 1 × 10^4^ in 0.1 mL of medium, in a 96 well plate. Natural extracts were diluted at maximum concentration of 3 mg/mL in DMSO. For each experiment were included, as negative controls, wells containing only cells and cells with 1% DMSO, both in 0.1 mL of medium; as background were considered wells containing only culture media; as treatments, were plated wells containing cells with extracts at the concentration of 5 and 30 µg/mL. As positive controls were used Cisplatin, MEK inhibitor, and doxorubicin, all at the concentration of 100 µM. All conditions were plated in duplicate and cells incubated were for 24 h.

### 4.5. Cytotoxicity Assays

For cell lines growing in adherence, the sulforodamine B (SRB) assay Kit (Abcam ab235935) was performed. After 24 h of treatment with compounds, cells were fixed with fixation solution for 1 h at 4 °C. After 3 washes in H_2_O, cells were stained with SRB solution for 15 min and rinsed with washing solution for 4 times. Protein-bound dye was solubilized with solubilization solution and the optical density was determined at 545 nm.

MTS Proliferation Assay Kit (Abcam, ab197010) was used for cells growing in suspension. 10 μL MTS (3-(4,5-dimethylthiazol-2-yl)-5-(3-carboxymethoxyphenyl)-2-(4-sulfophenyl)-2H-tetrazolium) was added to each well and incubated at 37 °C for 4 h. The absorbance was measured at 490 nm. For all the experiments, % of cytotoxicity was calculated as: [(O.D. vehicle) − (O.D. sample)/O.D. vehicle] * 100. Background correction was carried out by subtracting the O.D. of culture media.

### 4.6. D1 Cell Assay

D1 cells were plated on an untreated white flat 96-well plate at a density of 1.5 × 10^4^ cell/0.2 mL of complete culture medium and incubated with treatments for 24 h. Compounds were dissolved in MeOH at the maximum concentration of 0.3 mg/mL. Of this solution, 0.05 mL were used to perform the coating of the plate.

After 24 h, plates were centrifuged at 300 g for 3 min and washed with staining buffer (SB) (2% FBS; 0.1% sodium azide in PBS). Staining was performed with monoclonal antibody anti MHC-II APC, CD80 FITC, CD40 PE (REA custom mix from Miltenyi Biotech Auburn, CA, USA). Before acquisition, each sample was incubated with Propidium iodide solution (Invitrogen) for 10 min at room temperature.

### 4.7. MoDCs Assay

Human peripheral blood was obtained from discharge materials left during the manufacture to prepare blood bags to use for medical purposes and no identifying information on the donor was retained. The samples were recovered from different Transfusion Centers in the Campania Region (Italy), the blood donors were aware that leftover material could be occasionally used for various research purposes and they routinely gave their informed consent for this use when donating blood.

For each assay, human peripheral blood mononuclear cells were isolated from three healthy donors by routine Ficoll density gradient centrifugation. Monocytes were purified from human peripheral blood mononuclear cells using MACS CD14 microbeads (Miltenyi Biotech, Auburn, CA, USA) according to the manufacturer’s recommendation. Purity was checked by staining with a FITC-conjugated anti-CD14 antibody (Miltenyi Biotech, Auburn, CA, USA) and FACS analysis and was routinely found to be greater than 98%. Immature DCs were obtained by incubating monocytes at 1∙10^6^/mL in RPMI 1640 medium supplemented with 10% fetal calf serum, 1% L-glutamine 2 mM, 1% penicillin and streptomycin, human IL-4 (5 ng/mL), and human GM-CSF (100 ng/mL) for five days.

After five days in culture, MoDCs were incubated with natural compounds dissolved in the same conditions of D1 assay, in flat 96-wells at a concentration of 1.5 × 10^4^/0.2 mL. Stimulation with Pam2CSK4 1 µg/mL (Invivogen, San Diego, CA, USA) was used as a positive control. Cells treated only with vehicle (MeOH) were used as the control. Surface staining after 24 h was performed with HLA-DR FITC, CD83 PE and CD86 APC (Milteny Biotech, Auburn, CA, USA) REA monoclonal antibodies.

### 4.8. Diatom Cultivation

*Thalassiosira weissflogii* was cultured in 10 L photobioreactors filled with 0.22 μm FSW enriched with f/2 medium. Cultures were gently bubbled with filtered ambient air and grown in a climate chamber at 20 °C under 12 h:12 h light:dark cycle (100 μmol photons·m^2^·s^−1^). Cells were harvested at the stationary phase by centrifugation at 4000× *g* for 10 min at 4 °C in a swing-out rotor.

### 4.9. Extraction and Fractionation of T.weissflogii

Organic extracts were prepared and fractionated according to our previous protocol [50]. Briefly, methanol extracts (Ext) were obtained from wet pellet (about 200 mg) and fractionated (about 20 mg) on HR-X column (6 mL/500 mg) (Chromabond^®^ HR-X, Düren, Germany) by using five eluents, i.e., A, H_2_O 100%; B, MeOH/ H_2_O 1:1, C, ACN/ H_2_O 7:3; D, ACN 100% and E, CH_2_Cl_2_/MeOH 9:1. Fraction A was excluded from the assays, since it contains almost salts. An aliquot of 150 µg of organic samples was prepared for each bioassay. Fraction containing sulfoquinovosides was further purified by radial silica chromatography on Chromatotron (T-Squared Technology Inc, San Bruno, CA, USA) by a gradient of CHCl_3_: MeOH and followed by CHCl_3_: MeOH: H_2_O, 65:25:4 v:v:v.

The active fractions C were characterized by 1H NMR (600MHz).

### 4.10. Statistical Analysis

All data were analyzed by one-way ANOVA followed by the Tukey test for a multiple comparison test. A *p*-value less than 0.05 was considered as statistically significant. All analyses were performed using the GraphPad Prism 8.00 for Windows software (GraphPad Software, San Diego, CA, USA).

## Figures and Tables

**Figure 1 marinedrugs-18-00604-f001:**
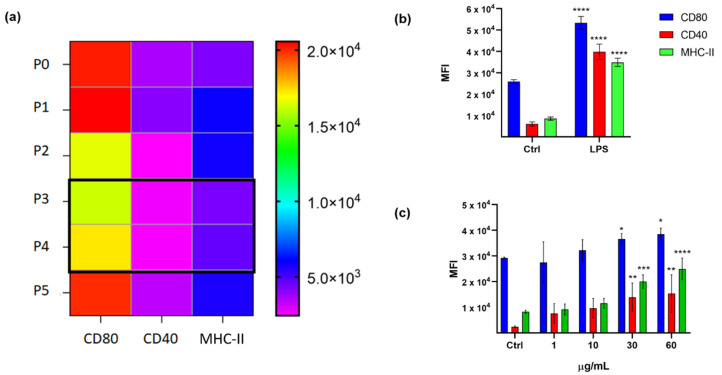
(**a**) Mouse Dendritic cell line (D1) surface marker analysis of CD80, CD40, and MHC-II in each cell passage along 12 days (*n* = 6) from P0 to P5. The color bar on the right shows the MFI (mean fluorescence intensity) measured for each marker; (**b**) surface marker expression analysis of D1 untreated (Ctrl) and treated with LPS (10 µg/mL; 24 h) (*n* = 5); error bars indicate standard deviations; (**c**) MFI of CD80, CD40, and MHC-II in D1 cells treated with Sulfavant A compared with cells treated with vehicle (Ctrl = MeOH) (*n* = 7); error bars indicate standard deviations; asterisks indicate significant differences from Ctrl; * *p* < 0.5, ** *p* < 0.01, *** *p* < 0.001, **** *p* < 0.0001.

**Figure 2 marinedrugs-18-00604-f002:**
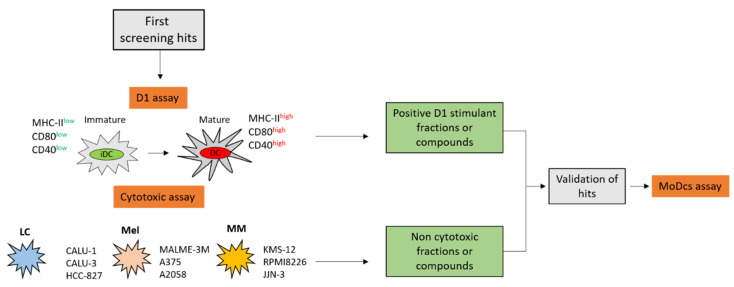
Overall scheme for hits selection. In the preliminary screening, natural products are tested on dendritic cell lines (D1) and for cytotoxicity on target lung carcinoma (LC), melanoma (Mel), and multiple myeloma (MM) cell lines. In the second step, the “candidate” samples are validated for immunomodulatory activity on human monocyte-derived dendritic cells (MoDCs).

**Figure 3 marinedrugs-18-00604-f003:**
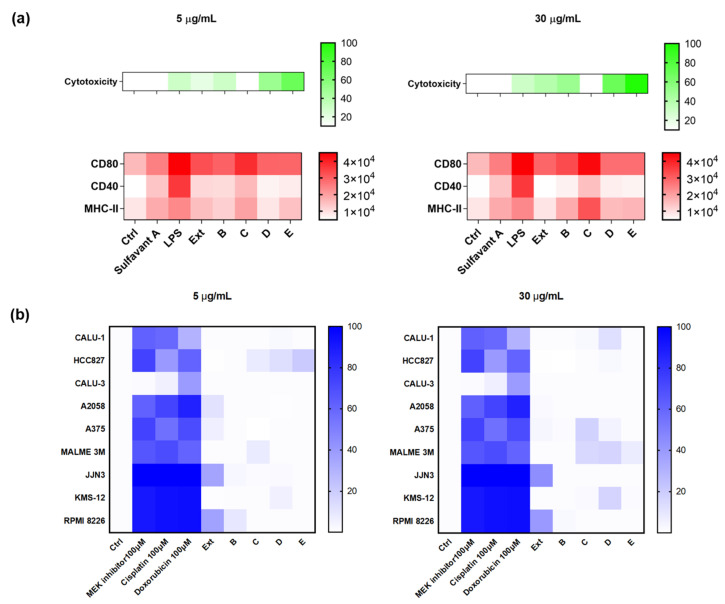
(**a**) Surface expression analysis of CD80, CD40, and MHC-II and percentage of vitality on D1 treated with the extract (Ext) and four fractions (B, C, D, E) that were derived from *T. weissflogii* at 5 and 30 µg/mL; all data were compared to the cells treated only with the vehicle (Ctrl) and cells treated with positive controls (LPS, Sulfavant A); (**b**) heat map of cytotoxicity assays conducted on the nine different cell lines. Cells were treated with positive controls, the extract (Ext) and four fractions (B, C, D, E) were derived from the marine diatom. Values reported in the color bar legend on the right indicate the % of cytotoxicity. The color bar on the right shows the MFI (mean fluorescence intensity) measured for each marker and the % of cytotoxicity.

**Figure 4 marinedrugs-18-00604-f004:**
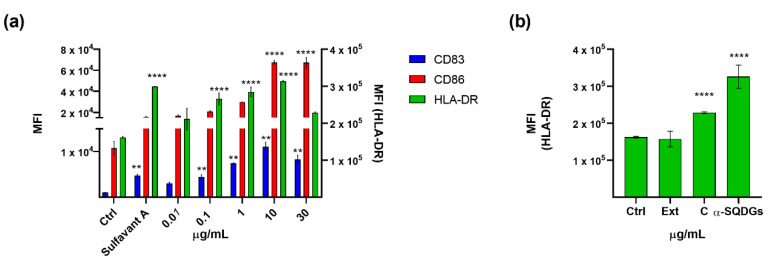
(**a**) Surface expression analysis of CD83, CD86, and HLA-DR in MoDCs treated with the fraction C derived from the extract of *Thalassiosira weissflogii* from 0.01 to 30 µg/mL; all data were compared to the cells treated only with the vehicle (Ctrl) and cells treated with Sulfavant A (30 µg/mL); (**b**) HLA-DR analysis of MoDCs treated with extract (Ext), SPE C fraction and purified α-SQDGs at concentration of 30 µg/mL; error bars indicate standard deviations; ** *p* < 0.01, **** *p* < 0.0001.

**Figure 5 marinedrugs-18-00604-f005:**
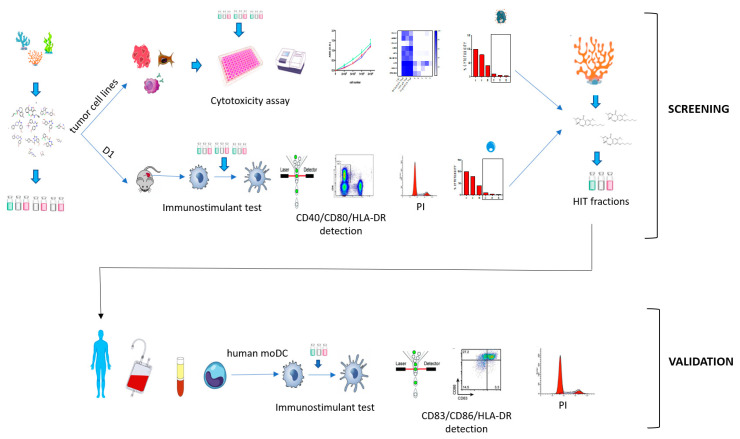
Screening platform workflow.

**Table 1 marinedrugs-18-00604-t001:** Tumor cell lines characteristics.

Cell Line	Mutations	Disease
**LUNG**		
**HCC827**	EGFR	Adenocarcinoma
**CALU-1**	KRAS; P53	grade III, epidermoid carcinoma
**CALU-3**	P53	Adenocarcinoma
**MELANOMA**		
**A375**	BRAF; CDKN2A; TERT	malignant melanoma
**A2058**	BRAF; TERT; TP53	Melanoma
**MALME-3M**	BRAF; CDKN2A; TERT	malignant melanoma
**MYELOMA**		
**KMS-12**	TP53	multiple myeloma
**RPMI8226**	EGFR; KRAS; TP53	Plasmacytoma
**JJN-3**	-	plasma cell leukemia

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
