# Peer review of "A New Bioassay Platform Design for the Discovery of Small Molecules with Anticancer Immunotherapeutic Activity"

_marinedrugs, 2020, doi:10.3390/md18120604_

Round 1

Reviewer 1 Report

In the present study, the authors searched a screening platform to select natural products with potential application in cancer immunotherapy. I think that the topic of this study is interesting. However, there are several sever flaws in this study. The most serious issue is that the authors analyzed the cell surface HLA-DR expression on mouse spleen D1 cells. Why do the mouse D1 cells express human antigen HLA-DR? This makes me suspicious of the present results.

Other comments:

  1. In the present study, human peripheral blood was used. However, there is no statement about the ethical approvement and informed consent. The authors have to describe it.
  2. In Line 358-359, it is described that two healthy donors were applied for the present study. However, in Line 392, there is a description that MoDCs were prepared from three donors. Which is true?
  3. There are some careless mistakes such as “24nhours” in Line 3308. The authors have to check the manuscript carefully. In addition, full name of some abbreviations such as TLR is missing.

Reviewer 2 Report

Comments to the authors

A new bioassay platform design for the discovery of lead compounds with anticancer and immunomodulant activity, Marine_Drugs_993043

The manuscript from Gallo et al. reports a novel bioassay to screen the immunomodulatory activity of compounds inducing dendritic cell maturation of immature murine D1 cells and activates human monocyte-derived dendritic cells.

The authors focus on a scientifically important field. Structure of the paper is logic and well organized. Due to the high mortality rate of solid malignancies in adults and the inventory of marine drugs as potential immunomodulatory agents, the manuscript fits in the scope of Marine Drugs journal.

I have one major comment:

  1. The screening is based on the upregulation of co-stimulatory molecules such as CD80/CD40 for D1 cells and CD83/CD86 for MoDCs with the induction of HLA-DR. The authors did not perform functional assay with the priming of T-cells using these activated DCs. However, it is emphasized in the Discussion that is one of the further aims. Also, the cytokine profile e.g. the expression of IL-12, IL-6 and IFN-g suggested to be monitored by the Reviewer in these D1 and MoDCs (ELISA, ELISPOT, LUMINEX or qRT-PCR). Alternatively, one experiment depending on the choice of the authors should be performed to demonstrate the immunostimulatory activity of activated D1 cells or MoDcs in a functional assay in vitro.

I have the following minor comments:

  1. The link between inflammation and Cancer has long been known. The ‘Introduction’ section is well organized, the mentioning of the effect of cancer-related inflammation on tumor progression in the paragraph 47-54 lines is missing. The inflammatory tumor microenvironment shape tumor immunity and render APCs unable to mount an affective priming for the adaptive arm. (seminal works from A. Mantovani, A. Sica or F. Balkwill).
  2. The XTT assay is mentioned in the Materials and Methods but not in the corresponding test below Table 1.
  3. The authors should specify that cell lines in the 249 line are myeloid cells (THP-1is immature monocyte, HL-60 is immature myeloid leukemia etc) used in the DC field but not DC cells.
  4. The Introduction or Discussion may be broadened with a Review about small molecule-based immunomodulators, (eg. Szebeni et al Pro-Tumoral Inflammatory Myeloid Cells as Emerging Therapeutic Targets, IJMS, https://doi.org/10.3390/ijms17111958) or similar paper.

Round 2

Reviewer 1 Report

Although the authors responded my comments, I am not satisfied with the authors’ response.

  1. First, I cannot believe the authors’ response to their mistake without the evidences. The authors have to add the information about catalog number of the antibodies used in this study. In addition, the authors have to show cytograms (or histograms) of flowcytometry data as supplementary data.
  2. I think that this study is not approved by the ethics committee. According to the “Instructions for authors”, the research using human materials needs an approval from an ethics committee before undertaking the research. Therefore, in this point, I think that this study is not suitable for the publication in the Marine Drugs. 

Reviewer 2 Report

The manuscript has been improved and the revised version can be accepted for publication.